# Bergmann’s Rule under Rocks: Testing the Influence of Latitude and Temperature on a Chiton from Mexican Marine Ecoregions

**DOI:** 10.3390/biology12060766

**Published:** 2023-05-24

**Authors:** Raquel Hernández-P, Hugo A. Benítez, Claudia Patricia Ornelas-García, Margarita Correa, Manuel J. Suazo, Daniel Piñero

**Affiliations:** 1Posgrado en Ciencias Biológicas, Unidad de Posgrado, Universidad Nacional Autónoma de México, Ciudad de México 04510, Mexico; 2Departamento de Ecología Evolutiva, Instituto de Ecología, Universidad Nacional Autónoma de México, Ciudad de México 04510, Mexico; 3Centro de Investigación de Estudios Avanzados del Maule, Instituto Milenio Biodiversidad de Ecosistemas Antárticos y Subantárticos (BASE), Universidad Católica del Maule, Talca 3466706, Chile; hbenitez@ucm.cl (H.A.B.); mcorreag@ucm.cl (M.C.); 4Centro de Investigación en Recursos Naturales y Sustentabilidad (CIRENYS), Universidad Bernardo O’Higgins, Avenida Viel 1497, Santiago 8370993, Chile; 5Colección Nacional de Peces, Departamento de Zoología, Instituto de Biología, Universidad Nacional Autónoma de México, Ciudad de México 04510, Mexico; patricia.ornelas.g@ib.unam.mx; 6Instituto de Alta Investigación, CEDENNA, Universidad de Tarapacá, Casilla 7D, Arica 1000000, Chile; suazo.mj@gmail.com

**Keywords:** chitons, size, shape, ecoregion, temperature, Bergmann’s rule

## Abstract

**Simple Summary:**

Chitons are a charismatic group of mollusks that live underneath rocks on shores throughout the world, where they experience variations due to environmental factors. Temperature may cause animals to grow to a larger size in colder climates, a trend originally ascribed to endothermic animals. In this work, advanced techniques were used to describe both shape and size of chitons living under different temperature regimes on Pacific shores. The results show that chitons are larger in colder zones, as occurs in endothermic vertebrates. Respiration, reproduction and/or feeding strategies may explain this trend in chitons.

**Abstract:**

Bergmann’s rule relates the trend of increasing body size with higher latitudes, where colder climates are found. In the Mexican Pacific, three marine ecoregions are distinguishable across a latitudinal gradient. *Stenoplax limaciformis* is an abundant chiton species that is distributed on rocky shores in these ecoregions. Geometric morphometric analyses were performed to describe the shape and size variation of *S. limaciformis* between marine ecoregions that vary in sea surface temperature with latitude, thus testing Bergmann’s rule. Individuals’ body shape ranged from elongated to wide bodies. Although there was variation in chitons’ body shape and size, the was no evidence of allometry among localities. The Gulf of California is the northernmost ecoregion evaluated in this work, where larger chitons were observed and lower sea surface temperature values were registered. The results suggest that *S. limaciformis* follows a trend to Bergmann’s rule, such as endotherms. These mollusks do not need heat dissipation, but they do need to retain moisture. In addition, larger chitons were observed in zones with high primary productivity, suggesting that chitons do not delay their maturation due to food shortage.

## 1. Introduction

The North Pacific is the largest ocean basin in the world with a wide variety of complex habitat patches and a high species diversity [1]. This has been the result of diverse types of geological processes, oceanographic factors, and climate regimes [2]. The oceans have been classified into biogeographic regions based on their degree of endemism [3,4], considering pelagic systems [5], or coastal and shelf waters [6]. On the other hand, the classification of marine ecoregions has been based on oceanographic factors such as bathymetry, the existence of large water masses, and marine currents, as well as benthic and pelagic species composition [7,8]. In the Northeastern Pacific, three marine ecoregions are distinguishable: the Mexican Pacific Transition (MPT), the Gulf of California [8] and the Southern Gulf of California [9].

The MPT is a fairly complex region, with a narrow shelf that drops abruptly close to the coast. The MPT extends from the coast of Jalisco to Oaxaca in Mexico and includes the southernmost tip of Baja California Sur. This ecoregion comprises high geographical heterogeneity, with numerous hills and seamounts, including a rift system and volcanic cones. The MPT is an ecoregion of high productivity and warm sea temperature (25–28 °C in winter and over 29 °C in summer), that is seasonally affected by different currents. During the winter, it is influenced from the north by the Northern California Current and from the south by the North Equatorial Countercurrent and the Costa Rica Coastal Current [8].

The Gulf of California (GC) ecoregion reaches from northern Sonora to the southwestern coast of the Baja California Peninsula and the Gulf islands. It is a semi-enclosed subtropical sea, which has a high seasonal variation characterized by its rich biodiversity, high biological productivity, and high rates of endemism of marine life [10]. Marine records suggest that its formation could be as ancient as the western movement of the Baja California Peninsula, with an estimated age of ten million years, formed during the Late Miocene and Early Pliocene [11]. Thus, the fauna that inhabits the Gulf of California arrived there from Central and South America, the Caribbean Sea (before the formation of the Panama Isthmus), and even across the vast stretch of the Pacific Ocean from the tropical Indo-West Pacific [11,12]. Finally, the southern zone of the Gulf of California is recognized as a distinct ecoregion, based on chiton species, named the Southern Gulf of California (SGC). That zone is a complex mix of waters and oceanic conditions that marks the northern distribution limit of a large number of chiton species that are widely distributed throughout the Panamanian realm [9].

Chitons belong to the class Polyplacophora of the phylum Mollusca. They are marine organisms that live on hard substrates, mainly rocks and coral rubble. Approximately 900 chiton species have been described in the world [13,14]. These mollusks have a shell divided into eight plates, which are overlapping and surrounded by a muscular girdle. The first and eighth plates are semicircular in shape, while the intermediate plates (second to seventh) have a “butterfly” shape. The eight plates that form the chiton shell allow locomotion and provide protection (e.g., rolling into a ball-like conformation), as well as some degree of flexibility over uneven and rough surfaces [15,16]. The surface of the dorsal plates of chitons’ shells are covered with aesthetes, which are sensory structures that have photosensitive, mechanoreceptor, chemoreceptor, or secretory cells, or even ocelli [14].

The genus *Stenoplax* is recognized by its elongated body [17]. Within the genus, the species *Stenoplax limaciformis* (G. B. Sowerby I, 1832) shows a wide latitudinal distribution from the Gulf of California up to the West American Central along the shallow rocky shores. It is less abundant toward the northernmost extreme of its range, and becomes increasingly common southward [18,19]. This species has variable color and sculpting patterns on its shells. It reaches >3 cm in length, inhabits shallow waters between 1–10 m deep and lives near crustose (brown and red) and filamentous (green) algae [17,20].

Climatic conditions such as temperature, UV radiation, precipitation and weather in general can impose physiological constraints on species. These conditions are highly variable over both space and time, so it is possible to observe their effects in species with wide geographic distributions [10,21]. A widely demonstrated ecogeographical rule is the one Carl Bergmann published in 1847, which predicts increasing body sizes of animals due to decreasing environmental temperature when approaching the poles [22]. In his 1847 work, Bergmann described the trend of increasing body size with latitudinal or altitudinal distance, with species exhibiting a smaller body size in warmer environments and larger body size in colder climates, which was coined “Bergmann’s rule” [22]. This rule is considered a thermoregulatory rule, as it predicts that animals will retain heat better with a higher ratio of volume to surface area [23]. For example, 72% of birds and 65% of mammal species studied followed this rule, leading to the conclusion that it is likely a valid ecological generalization for these two groups [24]. In ectotherms, however, it has been proposed that the opposite trend—“Converse Bergmann’s Rule”—may occur [25,26]. Ectothermic organisms obtain heat by absorbing it from the environment [27], so Bergmann’s rule has found mixed support in these species.

In an analysis of freshwater fishes, ~30% (5/18) species exhibited a positive trend (i.e., followed Bergmann’s rule) [28]. Turtles (chelonians) generally follow Bergmann’s rule, while lizards and snakes (squamates) exhibit the converse relationship to this rule, decreasing in size as latitude increases [29]. Support in invertebrates is also variable. Within insects, the butterfly *Auca coctei* follows the converse Bergmann’s rule [30], while the orthopteran *Trilophidia annulata* showed a concordant pattern with Bergmann’s rule [31]. For some marine invertebrates this rule has been tested, with patterns following this rule, such as the fiddler crab [32] and marine copepods [33]. In other groups such as marine bivalves, the relationship between body size and latitude across families with functional groups is still not clear, since the main variations occur between hemispheres and coastlines [34]. In chiton species from the southeastern Pacific, it has been observed that body size increases with latitude, following Bergmann’s rule both within and among species [35].

Adaptation to local geographic factors and long-term evolutionary diversification also produce differences in shape among individuals or their body parts [36]. Geometric morphometrics is the statistical analysis of shape variation and its covariation with other variables. These analyses can be visualized as shape and interpreted anatomically through Procrustes superimposition or shape-size space, which eliminates the effects of rotation and size [37,38]. The influence of size on shape is referred to as allometry in geometric morphometrics context [39]. In this approach, the measure of centroid size—the square root of the sum of squared distances of all landmarks of an object from their center of gravity—is used [39,40]. Geometric morphometrics describe shapes in detail, which also allows appropriate visualization and interpretation of the results, which is an advantage over traditional lineal morphometric measurements [36,41].

In this work, we aimed to explore the shape and size variation of *S. limaciformis* along a latitudinal gradient, where sea surface temperature is variable over the year. In addition, we use geometric morphometric tools and evaluate their utility in explaining Bergmann’s Rule and exploring the evolutionary response of the effect of temperature on the body size of chitons. To this end, sea surface temperature records were analyzed along ~2800 lineal kilometers in three marine ecoregions from Mexican portion of the Pacific Ocean where *S. limaciformis* is distributed.

## 2. Materials and Methods

The sampling was carried out at fifteen localities at different latitudes (Appendix A) in the three marine ecoregions: Gulf of California (GC), Southern Gulf of California (SGC), and Mexican Pacific Transition (MPT), located in the Mexican portion of the Pacific Ocean (Figure 1A). Samplings were conducted in bays or semi-enclosed zones at different times of the year (2019–2022), during low tide when possible. In each locality, three persons (each was always coordinated by the same person) examined movable rocks haphazardly for three hours, using a scraper to collect 5–15 individuals of *S. limaciformis* while snorkeling/diving in subtidal and intertidal zones during high tide or walking during low tide in some localities (Appendix A; Figure 1B–D). All specimens were transported in seawater and subsequently anesthetized by submerging in 25 mM MgCl_2_ solution. Chitons were preserved in an initial solution of 50%, 70% and finally 96% ethanol, with each individual carefully arranged into a dorsoventral position. The identification of *S. limaciformis* for all specimens was confirmed following the diagnosis suggested by Kaas & Van Belle (1987). *S. limaciformis* have radial areas with irregular, often beaded, concentric ridges; each ridge connects anteriorly to one of the longitudinal ribs on the central areas. However, some individuals have ribs formed by aligned nodules, while others show irregular uninterrupted ribs [20]. So, for the geometric morphometric analyses only adult specimens were used, which had valves sculptured with pustules or nodules towards the side margins on lateral areas of intermediate valves. Chitons with weak concentric ridges and without pustules, as well as individuals that were crooked, were discarded (Appendix A). Finally, 125 individuals from three ecoregions were used for the analyses: 34 from GC, 32 from SGC and 59 from MPT.

*S. limaciformis* body shape was quantified using a 2D geometric morphometrics approach. Pictures of each chiton were taken in dorsal view using a CANON EOS Rebel SL1 camera and using the same metric scale. Sixteen landmarks and eighteen semilandmarks (two curves; Figure 1E) were placed using *tpsDig2* v2.31 [42]. The semilandmarks were then transformed to landmarks because *geomorph* cannot read semilandmarks (append tps curve to landmarks option) using *tpsUtil* v1.78 [42]. The curves were defined again, using *define.sliders* (*geomorph*’s function), and in order to remove all mathematical influence of the shape, a generalized Procrustes analysis was performed. After this procedure, a covariance matrix of individuals was created to perform the following multivariate analyses (all analyses were performed between ecoregions). The shape space was visualized through Principal Component Analysis (PCA) and the first two components per marine ecoregion were plotted. Canonical Variate Analysis (CVA) was performed through 10,000 permutations of Mahalanobis distance from the pooled-within-ecoregions covariance matrix. The average shape of chitons per ecoregion was extracted, using the covariance matrix of the average shape variation to identify graphically the morphological variation between ecoregions. All analyses and plots were performed using R packages: *geomorph*, *Morpho*, *dplyr* and *ggplot2* [43,44,45,46]. Their size was quantified through centroid size, which was obtained in conjunction with Procrustes superimposition (Figure 1F). The centroid size for each ecoregion was plotted in a violin graph, using *ggplot2*. To calculate allometry between marine ecoregions, the shape scores were regressed onto centroid size. A Procrustes ANCOVA with 1000 permutations was performed, using *procD.lm* (*geomorph* function), with Procrustes distances among specimens, in order to describe patterns of shape variation as well as covariation between shape and other variables [47].

The environmental variable analyzed was the Sea Surface Temperature (SST). Satellite-based SST data were obtained from the National Oceanic and Atmospheric Administration (NOAA) server ERDDAP (Easier access to scientific data), by accessing the dataset *Sea Surface Temperature*, *Coral Reef Watch*, *CoralTemp*, *v3.1*—*Daily*, *1985-present*; dataset ID: CRW_sst_v3_1 [48,49]. Metadata were downloaded in .csv files for each degree latitude in the three marine ecoregions (Appendix A). SST data were averaged to determine the warmest and coldest trimesters. Trimesters were calculated following the method for Bioclimatic Predictors [21]: (1) averaging SST per month over 36 years (1985–2020), then (2) grouping into consecutive three-month intervals (e.g., January–February–March; February–March–April), which were then averaged to obtain 12 trimester values (Appendix A). Finally, the annual mean and the difference between the warmest and coldest trimesters (seasonal range) were calculated, using the previously described trimester values. These data were analyzed and plotted using R packages: *tidyverse* and *ggplot2* [43,50].

To evaluate Bergmann’s rule, we performed one regression of the centroid sizes of chitons (per individual) against the degree latitude of the sampling locality and a second regression of centroid size against the annual average of SST for each sampling locality. Subsequently, centroid size was regressed on the warmest, coldest and seasonal range of SST values. All variables were plotted by ecoregion using *ggplot2* [43].

## 3. Results

The first principal component contained 47.44% of the shape variation (Appendix A); however, the three marine ecoregions were not graphically distinguished. Chitons from the MPT and SGC were more similar to each other than to chitons from the GC. Geographically, the SGC lies between the beginning of the MPT and the entrance of the gulf, which is an enclosed area (Figure 1A); thus, the SGC is influenced by the MPT and GC ecoregions. In the CVA, the body shape of *S. limaciformis* differed among the three marine ecoregions, and the shape of those from the GC was the most distinct (Figure 2).

The extremes of the chiton shapes were noticeable between specimens from more distant marine ecoregions; some chitons from GC had a more elongated body shape compared to the other two ecoregions, and some chitons from MPT had a wider body shape (Figure 3C). However, these differences were not present in all specimens from each ecoregion, so it is not possible to describe chitons from an ecoregion with a specific shape.

Regarding the centroid size, some chitons from the GC were larger than specimens from the SGC and MPT (Figure 3A). Additionally, the centroid size of chitons from GC had higher variance than the other ecoregions (Figure 3B). GC is the ecoregion located at the highest latitude, and chitons from this ecoregion had a larger body size than the other two ecoregions (the SGC and MPT, which were similar to each other). The ANCOVA results further confirmed differences in shape and size among the ecoregions, which evidenced significant CS differences (*p* = 0.001) for shape and size independently. However, the allometry analysis that showed that the interaction between “Ecoregion*Centroid Size” was not significant (*p* = 0.211; Table 1).

This result means that although there are differences in shape among marine ecoregions, it is not associated with centroid size, so allometry is not significant between marine ecoregions of *S. limaciformis*. In the present study, the samples were located between 15° to 30° north latitude. In this range, the centroid size of *S. limaciformis* increases at higher latitudes (Figure 4A), where SST is lower (Figure 4B). Based on trimester values, the coolest trimester reached lower temperatures in high latitudes where chitons are larger in comparison to chitons from low latitudes. This relationship is supported by regression, which showed significant *p* values (0.006; Figure 4C). In the opposite scenario, higher temperatures are reached in lower latitudes. The warmest trimester reached nearly 30 °C; the smallest chitons occurred at these sites, and the regression had a significant *p*-value (0.0026; Figure 4D). The difference between minimum and maximum SST ranged from 2.5 to 12.5 °C in the three ecoregions. Longer chitons were obtained in sites where SST had ranges of 12.5 °C throughout the years (Figure 4E).

## 4. Discussion

The present study is a description of the shape and size variation of *S. limaciformis*. The morphology of chitons is peculiar because of their dorsoventrally flattened bodies and their eight plates assembled by a girdle. Chitons are often unnoticed because they live under rocks, and differences between specimens and species are not immediately apparent. Nevertheless, in this work, the geometric morphometrics approach allowed us to identify differences in body shape among marine ecoregions. This study is the first body shape description of a chiton across a latitudinal gradient at the intraspecific level. Although our results suggest a modest level of differentiation between ecoregions, and larger differences between the extreme ecoregions, the relationship is not completely linear, since the analyses describe a multivariate space of different shapes.

*S. limaciformis* has shape and size variation, and each shape (elongated or broad) is apparently independent of size; this result means that allometry is not present in this chiton. However, the considered size is centroid size, which was obtained from a multivariate space. In the absence of allometry, the only size related to shape is the centroid size [38]. The size of chitons was described using centroid size, and larger chitons were observed in higher latitudes where SST is lower, so this species follows Bergmann’s rule, as has been found in endotherms [25].

Existing hypotheses of body size evolution of ectotherms propose that under cool conditions, growth is slow and they mature at a larger body size than under warm conditions [51]. This pattern has been observed under laboratory conditions, related with the fast increase in metabolic oxygen consumption with temperature, in comparison to the rate of oxygen diffusion, so at high temperatures ectotherms can reduce the sizes of some cell types. Particularly, in aquatic environments oxygen solubility decreases when the temperature increases [52].

Size variation may also be due to the environmental variance in each ecoregion, and likewise may influence chiton biology such as feeding or respiration. Chitons have a foot with which they clamp themselves to hard surfaces, which is surrounded by a mantle groove containing multiple ctenidia arranged serially along each side of the foot. The number of ctenidia and their size can differ between individuals [16], and they also increase during chiton growth [53]. Water currents pass from anterior to posterior through the mantle groove that is covered externally by a fleshy girdle [16]. Tidal exposure levels may influence the foot size of chitons; a larger foot may allow a better grip to the substrate, providing protection against predators and possible desiccation. In addition, the number of ctenidia may increase in longer individuals, allowing them to continue gas exchange even during low tide. Likewise, in some species of coastal chitons, the smaller individuals are found in crevices and hollows that are protected from waves and the sun, while larger ones may be found in exposed zones [54]. One of Bergmann’s original ideas was that body size is related to both heat production and heat dissipation. This process is due to oxygen bonding [55]. Although chitons are ectotherms and they do not maintain a constant body temperature, they should maintain constant moisture.

A relationship between larger chitons and their diet breadth has been observed in *Acanthopleura echinata*, a chiton that is found in the southeastern Pacific. Dietary breadth also increases with latitude, but it is not related to upwelling nor to SST [56]. In this work, larger chitons were observed at GC, where upwellings occur and lower SST have been recorded. Upwellings have high primary productivity [57]. Additionally, the intensity of metabolism increases with body size, increasing the requirement for the amount and constancy of food [58]. A high-productivity environment, such as that of the GC, means that chitons and other herbivores have constant access to food. Although factors such as moisture content or primary production are to a high degree correlated with temperature, it is not always possible to determine their individual contribution to Bergmann’s clines, which is not universally supported in ectotherms [58].

Another factor that may contribute to Bergmann’s rule is reproduction [58]. Along with species distribution, the organisms in a warmer climate (where resources are constant along seasons) spend less time growing, reaching the reproduction stage earlier. Meanwhile, organisms in colder climates delay their reproduction because they allocate resources to growth [58,59]. In *Chiton articulatus* it has been observed that during the reproductive stage, somatic indices diminish considerably during gonad development, especially when the SST reaches maximum values [60]. In the current study, we found larger chitons at lower SST. Although the reproductive season of *S. limaciformis* is unknown, there are relatively broad seasonal changes in SST, such that it could potentially reproduce under a range of temperature conditions. On the other hand, slower growth increases the risk of death for organisms before they reach the maturity stages [61]. However, it is known that *Acanthopleura* and *Chiton* from the Caribbean Sea reach sexual maturity by the end of the first year, resulting in overlapping generations of reproductive adults (i.e., first-year and older adults reproduce simultaneously) [62].

Geomorphological features of sites can be possible causes of latitudinal variation of size for these mollusks. For example, the Gulf of California is the most recent ecoregion analyzed in this work, which may imply that its populations have a higher size variation in comparison with other populations that have been connected for a longer time. Secondly, the Northern Gulf of California has a very shallow shelf (depth < 30 m) and a slight slope, resulting in high tide amplitudes [8,57], such that benthic zones are exposed during low tide. Although chitons remain in tide pools in that time, larger chitons with more ctenidia could be better at withstanding the tidal change. On the other hand, MPT has a steeply sloping shelf [8]. Thus, chitons living in these zones remain submerged even during low tide, and they can breathe and maintain their moisture constantly, regardless of their number of ctenidia or body size. In addition, GC is a seasonally dynamic sea, dominated by the Pacific Ocean. During the summer the currents are caused by coastal trapped waves, and during winter they are wind-driven. These changes affect local processes, such as the altering of thermohaline characteristics of the upper layer waters, which are important at a shorter time scale [57]. Seasonal changes in temperature throughout the year influence the growth of organisms, while increasing temperatures can cause a decrease in the sizes of emerging adults [63]. Finally, at the entrance of the Gulf of California, which corresponds to the SGC ecoregion, latitudinal displacements of the equatorial current system occur, which determine how far south the California Current (cold water) will flow and how far north the Costa Rican Coastal Current (warm water) will flow [57]. These temperature variations coincide with what was observed in this work—larger chitons in higher latitudes (GC ecoregion) where the lower temperatures and broader range of SST were obtained, and smaller individuals in lower latitudes at higher temperatures.

Ecological and environmental variables change with latitude, while some ecological characteristics such as feeding or substrate may influence body size [64,65]. We can therefore expect that ecologically similar families exhibit similar trends. However, families of bivalves with similar trends are ecologically diverse, so the variation between the observed regions in these mollusks has been driven by diverse processes, past and present [34]. Based on the results of this work, the relationship between latitude and body size of each chiton species cannot be predicted even if they have the same ecological niche.

One idea of Bergmann’s rule is that “if we could find two species of animals (endothermic) which would only differ from each other with respect to size, the geographical distribution of the two species would have to be determined by their size” [22]. *S. limaciformis* is considered in a group of sibling species with *S. purpurascens* and *S. floridana*, which are distributed in the Caribbean Sea. *Stenoplax limaciformis*, *S. purpurascens*, and *S. floridiana* are reproductively isolated from each other, and although they are superficially nearly morphologically identical, they do show small but consistent morphological differences [66]. If the large size of *S. limaciformis* from high latitudes is due to temperature, one would expect *S. purpurascens* and *S. floridana* from the (warmer) Caribbean Sea to have smaller sizes.

## 5. Conclusions

Variations in chiton body shape were observed between marine ecoregions, with larger chitons found at higher latitudes where SST is cooler. Thus, *S. limaciformis* follows Bergmann’s rule. Geometric morphometrics is an adequate tool to describe chiton shape, which is variable but does not relate to size (i.e., there is no allometry). A relationship between body size and temperature was observed. However, it is paramount to include data on other oceanographic factors such as dissolved oxygen or chlorophyll a in order to better describe this and other chiton species. Since chitons adhere to the substrate, the size and shape of their bodies affect individuals’ access to resources. Thus, this work, describing variation in body size and shape across populations lays the foundations for future work on physiology, genomics or other fields in chitons or even other invertebrates.

## Figures and Tables

**Figure 1 biology-12-00766-f001:**
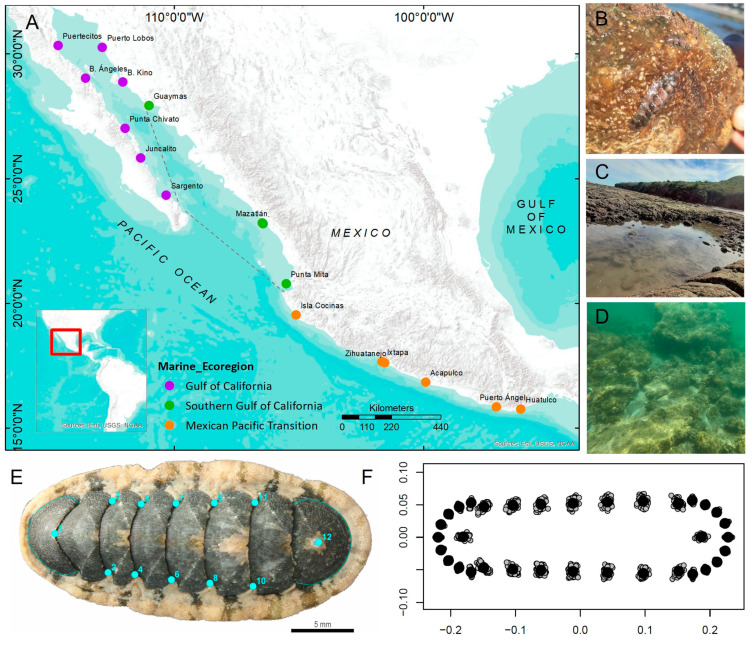
(**A**) Geographic location of sites where *S. limaciformis* specimens were obtained. Dotted lines indicate separation between ecoregions; (**B**) *S. limaciformis* adhered to a rock; (**C**) Mazatlán, a site from the Southern Gulf of California, during low tide; (**D**) Huatulco, a site from the Mexican Pacific Transitional that always remains submerged; (**E**) Location of landmarks and semilandmarks (curves) on *S. limaciformis*; and (**F**) Procrustes superimposition of *S. limaciformis*.

**Figure 2 biology-12-00766-f002:**
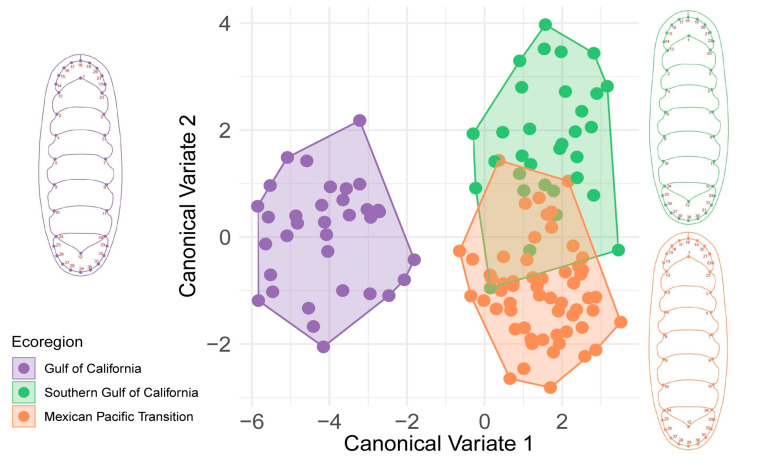
Shape description. Canonical variate analysis of *S. limaciformis*. Each color represents a marine ecoregion. Mean shape is displayed in the same color as the corresponding marine ecoregion.

**Figure 3 biology-12-00766-f003:**
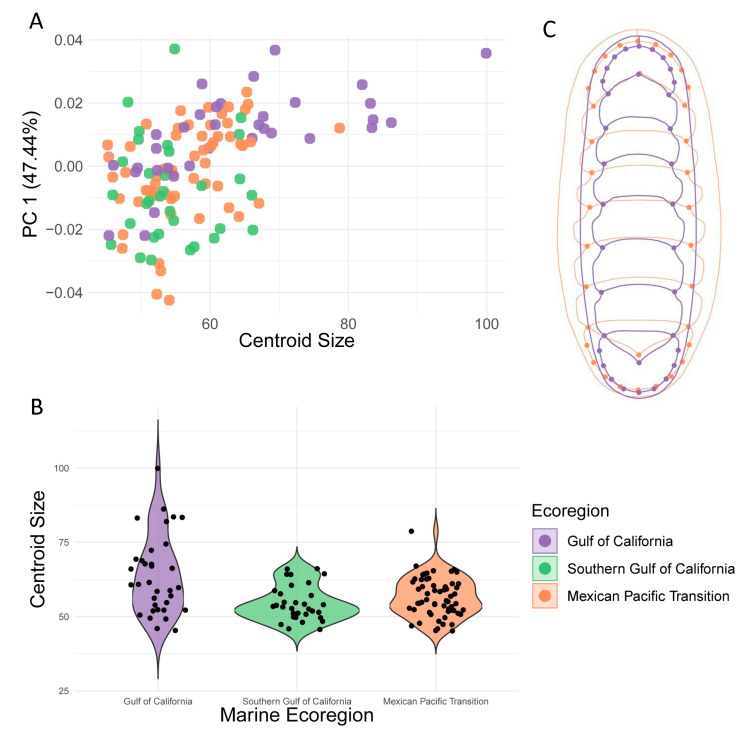
Allometry. (**A**) Shape variation accumulated in principal component 1 scores due to centroid size of *S. limaciformis*. (**B**) Centroid size variation in each marine ecoregion. (**C**) Chiton shape based on extreme values of principal component 1 scores, from Gulf of California and Mexican Pacific Transition.

**Figure 4 biology-12-00766-f004:**
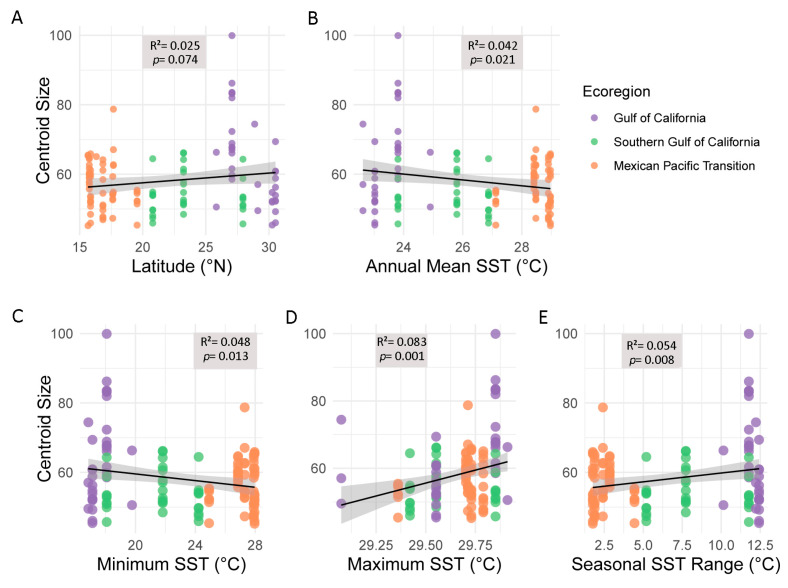
Bergmann’s rule. (**A**) Regression of centroid size on latitude per individual; (**B**) regression of centroid size on sea surface temperature per individual; (**C**) regression of centroid size on minimum (trimester) sea surface temperature per individual; (**D**) regression of centroid size on maximum (trimester) sea surface temperature per individual; and (**E**) regression of centroid size on the range of sea surface temperature per individual.

**Table 1 biology-12-00766-t001:** Procrustes ANCOVA model of shape among marine ecoregions.

	SS	MS	F	*p*
Centroid size	0.0090	0.009	20.04	0.001 **
Ecoregion	0.0043	0.0021	4.84	0.001 **
Ecoregion * Centroid Size	0.0012	0.0006	1.33	0.211

SS—Sum of squares, MS—Mean sum of squares, ** significant *p* values.

## Data Availability

We will upload the data to Dryad.

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
