# Peer review of "Bergmann’s Rule under Rocks: Testing the Influence of Latitude and Temperature on a Chiton from Mexican Marine Ecoregions"

_biology, 2023, doi:10.3390/biology12060766_

Round 1

Reviewer 1 Report

In this study, chitons were collected from the intertidal zone (by snorkeling at high tide) but it is not indicated, at what intertidal zone they were collected. Were they collected from comparable zones on each shore? No comparison is provided of the tidal regime in each location, although the authors acknowledge that the northern Gulf of California has a large tidal amplitude and some benthic zones are exposed during low tide, whereas in areas with a less pronounced slope chitons may remain submerged even during low tide.

There is no population structure given for chitons on any of the shores. The 5–15 specimens that were taken from each shore, what size/age group did they represent? How were they selected? If size is being compared at different locations, then some sort of random collection is needed. Perhaps using a grid. Haphazard collection, if that is what was used, cannot not be statistically analysed. The authors need to describe exactly their sampling method. If 5–15 individuals were collected, then some (at least one) location must have consisted of just 5 individuals. This is not a statistically viable number. No table is provided to show the sample size at each location. It is essential that they describe clearly the sampling methodology and tabulate the number of individuals collected from each shore.

The manuscript examines the relationship of sea surface temperatures with body size and shape in an intertidal mollusc on some Pacific coasts. The temperatures upon which this study is based were sea surface temperatures obtained from satellite data. Chitons, however, do not live at the sea surface. The intertidal region of marine shores is bathed by ocean waters during high tide but during low tide it experiences atmospheric temperature fluctuations affected by variable mounts of diurnal insolation/nocturnal cooling and by circadian precipitation. In this respect, atmospheric temperature data may also be  important. The aspect of the shore (which is not described for any of the locations) may also be an important factor. No ground-truth data appear to have been collected on temperature minima, maxima and averages experienced by chitons in situ underneath the rocks where they actually live on the shores analysed. It has not been tested whether crude SST data actively reflect the real temperature variations existing under rocks where the chitons live. We cannot estimate whether SST and latitude are together a good (and simple) predictor of micro habitat temperatures where chitons exist on different shores under complex environmental influences, because no ground truth data are apparently provided. (I was not able to access any of the supplementary data on the internet)

The hypothesis that the size of chitons is directly related to temperature (Bergmann's rule) is rather weak. It has not been shown that SST is a good surrogate for the actual microhabitat temperatures experience by chitons. There are very many confounding factors that might affect chiton size: humidity, food availability, feeding time (emersion-immersion) frequency when low tides fall during the day or during the night and reproduction timing, none of which seem to have been analysed for a possible latitudinal correlation.

I would recommend that the manuscript should be corrected for language by a native English speaker. I do not have the time to correct the entire manuscript for it's English, but I attach, as an example, a version of the 'Simple Summary' corrected for the English Language.

Simple Summary: Chitons belong to a charismatic mollusc group that live underneath rocks on shores throughout the world, where they experience variations in environmetntal factors. Temperature may cause animals to grow to a larger size in colder climates, a trend originally ascribed to endothermic animals. In this work, advanced techniques were used to describe both shape and size in chitons, living under different temperature regimes on some Pacific shores. The results show that chitons are larger in colder zones as is the case in endothermic vertebrates. Respiration, reproduction and/or feeding strategy may explain this trend in chitons. 

Author Response

:D

Reviewer 2 Report

This paper analyzes the relationship of latitude and size and shape of chitons. Many analyses have been undertaken to test the hypothesis and some interesting points are included in the discussion part. However, there are several questions that need to be addressed. 1) Quite a few grammatical problems. 2) Sampling methods were not entirely presented. 3) The selection of sites in GC is not very appropriate. Therefore, it might need to be certified whether data in this area affect the final results. 4) Based on the discussion and the data used in the paper, there is a high possibility that the size-latitude pattern might be caused by local environmental factors, instead of following Bergmann’s rule. Detailed comments are marked in the attached PDF.

Quite a few grammatical problems exist in the paper, and some of them are underlined.

Author Response

:D

Round 2

Reviewer 1 Report

The manuscript has been significantly improved and is now acceptable for publication after a small addition has been made.

There is always going to be a problem for statistical analysis where haphazard sampling, as opposed to random sampling, has been carried out. In this study, where shores are sampled haphazardly, it was important to reduce the possibility of sampling bias by employing the same samplers on all shores. On line 146 it is stated that "In each locality, three persons examined moveable rocks haphazardly....." It should be clearly stated that these were always the same three persons who carried out the sampling. If this is not so, and different persons sampled different shores, then this needs to be stated as a contributor to possible sampling errors.

Just a thought, perhaps for future research, since chitons clamp themselves to rock surfaces and need to have a good fit with the surface to prevent dislodgement by predators or by wave action, could the form of the rock surface and its irregularities, play a role in setting limits to their size range on different shores with different geological bedrock?

Author Response

Point 1. There is always going to be a problem for statistical analysis where haphazard sampling, as opposed to random sampling, has been carried out. In this study, where shores are sampled haphazardly, it was important to reduce the possibility of sampling bias by employing the same samplers on all shores. On line 146 it is stated that "In each locality, three persons examined moveable rocks haphazardly....." It should be clearly stated that these were always the same three persons who carried out the sampling. If this is not so, and different persons sampled different shores, then this needs to be stated as a contributor to possible sampling errors.

 Response

Thank you for your comment. Some shores have a few movable rocks so, if we carried out a random sampling in these sites, we could reduce the number of chitons found. Each sampling locality was obtained by three different persons (See line 146). However, they always were coordinated by the same person (Raquel H-P). She was supported by another experienced person in chiton sampling and a third experienced person in marine invertebrate sampling. Although we didn’t describe it in the “material and methods” section, in the Gulf of California (Baja California Sur state) we obtained samples in two different sampling trips and explored other shores: Ensenada Blanca, Pichilingue, El Requeson, Playa La Concha, El Pulguero and Cabo Pulmo. However, in those sites, we didn’t find any chiton of S. limaciformis species. The number of chitons in GC is low, but it represents the abundance of populations in this ecoregion.

Point 2. Just a thought, perhaps for future research, since chitons clamp themselves to rock surfaces and need to have a good fit with the surface to prevent dislodgement by predators or by wave action, could the form of the rock surface and its irregularities, play a role in setting limits to their size range on different shores with different geological bedrock?

Response

Thanks, that is a great idea. We hope to test those effects in future research. 

Reviewer 2 Report

After a minor revision, the paper should be ready to be accepted.

A few grammatical problems existed.

Author Response

Thank you for your revision

We find a few details. See lines 20, 90, 377

Our manuscript has already undergone an extensive English revision